# Blurred-Dilated Method for Adversarial Attacks

**Yang Deng**
School of Software Engineering
Sun Yat-sen University
dengy73@mail2.sysu.edu.cn

**Weibin Wu**[*]
School of Software Engineering
Sun Yat-sen University
wuwb36@mail.sysu.edu.cn

**Jianping Zhang**
Department of Computer Science and Engineering
The Chinese University of Hong Kong
jpzhang@cse.cuhk.edu.hk

**Zibin Zheng**
School of Software Engineering
Sun Yat-sen University
zhzibin@mail.sysu.edu.cn

## Abstract

Deep neural networks (DNNs) are vulnerable to adversarial attacks, which lead to incorrect predictions. In black-box settings, transfer attacks can be conveniently used to generate adversarial examples. However, such examples tend to overfit the specific architecture and feature representations of the source model, resulting in poor attack performance against other target models. To overcome this drawback, we propose a novel model modification-based transfer attack: Blurred-Dilated method (BD) in this paper. In summary, BD works by reducing downsampling while introducing BlurPool and dilated convolutions in the source model. Then BD employs the modified source model to generate adversarial samples. We think that BD can more comprehensively preserve the feature information than the original source model. It thus enables more thorough destruction of the image features, which can improve the transferability of the generated adversarial samples. Extensive experiments on the ImageNet dataset show that adversarial examples generated by BD achieve significantly higher transferability than the state-of-the-art baselines. Besides, BD can be conveniently combined with existing black-box attack techniques to further improve their performance.

## 1   Introduction

In recent years, deep learning has achieved great success in the field of computer vision. Despite the shining achievement, deep neural networks (DNNs) are vulnerable to adversarial attacks [1, 2]. It is easy to deceive deep neural networks into making incorrect predictions by adding small adversarial perturbations to natural images, which are almost imperceptible to the human visual system [3]. This has brought significant security implications with the wide application of deep neural networks [4, 5]. Therefore, to enhance the robustness of deep neural networks, more and more researchers have begun to focus on adversarial attacks and defenses.

Adversarial attacks can be mainly divided into two types: white-box attacks and black-box attacks. In white-box attacks [2, 4, 6], attackers have access to all the information about the target model, such as the model architecture and parameters, so adversarial samples can be directly produced through gradients with a high success rate. However, white-box attacks are often difficult to implement in reality because attackers cannot obtain all the information of the target model. Black-box attacks can overcome this problem. Prevalent black-box attacks can be generally categorized as query-based and transfer-based. This paper mainly focuses on transfer-based black-box attacks, where attackers

---

[*]Corresponding author.

37th Conference on Neural Information Processing Systems (NeurIPS 2023).

utilize a substitute model to craft adversarial samples and apply these samples to attack black-box victim models [7, 8, 9].

However, existing transfer-based black-box attacks can overfit to the source model [10, 11]. As a result, the discrepancies between the substitute model and target models, which stem from their differences in architectures, parameters, and other aspects, may lead to poor attack performance of the generated adversarial examples against the victim models.

Therefore, numerous endeavors have been made to enhance the transferability of transfer-based attacks. The recent prominent techniques can be classified into three categories: gradient optimization, input augmentation, and model modification. Among them, the research direction that particularly interests us is model modification-based attacks. For example, the Intermediate-Level Attack (ILA) [12] and ILA++ [13] generate transferable adversarial examples by attacking intermediate feature maps. Skip Gradient Method (SGM) [14] utilizes skip connections to improve the transferability of adversarial samples. Linear Backpropagation (LinBP) [15] skips some non-linear activation layers during backpropagation. Although these attack techniques narrow the gap between the substitute model and the victim model, the transferability of the generated adversarial examples is still unsatisfactory. Additionally, they mostly focus on modifying the backpropagation of the source model while neglecting its forward propagation.

In this paper, we explore whether modifying the architecture of the source model can enhance the transferability of adversarial attacks. We propose a novel model modification-based transfer attack: Blurred-Dilated method (BD). In short, BD works by reducing downsampling operations while introducing BlurPool [16] and dilated convolutions [17] in the source model. Then BD employs the modified source model to generate adversarial samples. Extensive experiments on the ImageNet dataset confirm that BD can achieve significantly higher transferability than the state-of-the-art baselines.

In summary, our main contributions are as follows:

- We propose a novel model modification-based transfer attack: Blurred-Dilated method (BD). BD reduces downsampling operations and adds BlurPool and dilated convolutions in the original source model. It then employs the modified source model to generate adversarial samples.

- We conducted extensive experiments on the ImageNet dataset to validate the effectiveness of BD. Experimental results confirm that BD can consistently outperform the state-of-the-art benchmarks by a large margin.

- We show that the proposed BD can be conveniently combined with other transfer attacks to further improve their performance.

## 2   Related Work

To enhance the transferability of adversarial attacks, recent techniques can be categorized into three types: gradient optimization, input augmentation, and model modification.

**Gradient optimization.** The Fast Gradient Sign Method (FGSM) family is the mainstream method in this category. FGSM [2] perturbs a clean example to generate an adversarial example by adding a small perturbation along the direction of the gradient sign, using only one iteration. The Iterative Fast Gradient Sign Method (I-FGSM) [4] and the Projected Gradient Descent method (PGD) [6] are iterative versions of FGSM for constructing adversarial examples. In comparison to FGSM, which only takes a single step along the direction of the gradient sign, these iterative methods deliver better attack performance. Momentum Iterative Method (MI-FGSM) [18] integrates momentum into the gradient at each iteration, allowing previous gradients to guide current updates.

**Input augmentation.** We briefly introduce some popular transfer attacks in this category. Diverse Input Method (DIM) [19] applies various transformations to input images in each iteration (e.g., random resizing and padding). Scale-Invariant Method (SIM) [20] optimizes perturbations utilizing the input images' scaled copies. In addition, the Translation-Invariant Method (TIM) [21] applies a preset kernel to the gradients with respect to the input images.

**Model modification.** Due to overfitting to the source model that is used to generate adversarial samples, the current transfer-based attacks usually exhibit limited transferability. Some researchers attempt to enhance the substitute model to improve the attack success rates. Since the decision

boundaries of different DNNs are different, employing an ensemble of source models is the most direct way to achieve this goal [7]. The Ghost Network [22] applies random dropout and skip connection erosion to simulate a large number of source models. However, creating and employing an ensemble of source models is still challenging and computationally expensive. Therefore, more works tend to focus on only one substitute model. Intermediate-Level Attack (ILA) [12] and ILA++ [13] craft transferable adversarial examples by attacking intermediate feature maps. Skip Gradient Method (SGM) [14] discovered that the gradient flow through skip connections can improve the transferability of adversarial samples. Therefore, SGM introduces a decay factor to reduce the gradient of residual modules. Linear Backpropagation (LinBP) [15] skips some non-linear activation layers during backpropagation.

Different from previous model modification-based transfer attacks that mainly focus on backpropagation, our proposed BD considers both backpropagation and forward propagation. Extensive experiments on the ImageNet dataset verify that our proposed method can effectively narrow the gap between the substitute model and the victim models, and thus significantly improve the transferability of the generated adversarial samples. Compared with state-of-the-art model modification-based transfer attacks, our method can also achieve significantly better attack success rates.

## 3 Method

In this section, we provide a detailed introduction of our proposed technique. We first introduce our motivation in Section 3.1. Then in Section 3.2, we elucidate BlurPool [16] and dilated convolutions [17], which are the key components in our proposed method. After that, we explain the general strategy of our proposed BD in Section 3.3. Finally, we present the implementation details of BD with popular architectures as examples in Section 3.4.

### 3.1 Motivation

Based on our preliminary experiments, we observed that: (1) The adversarial samples generated by ResNet [23] usually have better transferability than those generated by other models. We think the reason may be that the skip connections of ResNet connect the lower-layer feature maps to the higher-layer feature maps, which helps to transfer more low-level features and reduce the information loss caused by downsampling. Due to the large resolutions of the raw input images, models need to conduct downsampling multiple times, resulting in a large amount of information loss. Besides, due to the structural differences between models, the discarded features are different. Therefore, preserving more low-level features, as ResNet does, can improve adversarial transferability. (2) On CIFAR-10/100 [24], the adversarial transferability between models is usually better than that on ImageNet [25, 15]. Since the resolutions of the images on CIFAR-10/100 are smaller, there are fewer downsampling in the CIFAR-10/100 models. Therefore, different CIFAR-10/100 models all maintain more common low-level features, making the generated adversarial sample easy to transfer.

Given the above observations, we think that employing a source model that can preserve more low-level features of an image can help to improve adversarial transferability. Such a source model can guide adversarial samples to more thoroughly and precisely destroy the low-level features of a clean image. As a result, the generated adversarial samples are more transferable, since different models all use low-level features to extract high-level semantics and then make predictions. Although downsampling is widely used in various neural networks [26, 23, 27, 28], downsampling will discard feature information [29], which will reduce the details of an image (i.e., the low-level features) during forward propagation and thus hinder the transferability of the generated adversarial examples. Therefore, we propose our Blurred-Dilated method (BD) to modify the source model to preserve more low-level features of an image. As a result, the modified source model can generate more transferable adversarial samples.

### 3.2 BlurPool and Dilated Convolution

To preserve more low-level features of an image, we need to reduce downsampling in the original source models. To this end, we need to add BlurPool and dilated convolutions to the original source models.

**BlurPool.** In convolutional neural networks (CNNs), pooling layers or ordinary convolutions with strides greater than 2 would conduct downsampling on the input feature map to reduce the size of the input feature maps and increase the model's receptive field. However, since some features are discarded in the downsampling process, information may be lost. Traditional pooling aggregates all pixels within the window to a single value. In contrast, BlurPool aims to minimize information loss caused by pooling operations via replacing the traditional pooling kernel with a Gaussian kernel. This method enables more detailed weighted averaging over features within the pooling window, while improving shift-invariance. Specifically, when using a Gaussian kernel (we adopt 4×4 to follow the previous work [16]), BlurPool applies a Gaussian-weighted function to the neighborhood area around each feature for convolution calculation, and obtains a weighted average result. To further reduce the result's feature map spatial size, common downsampling strategies (such as using a large stride) are typically applied. Compared to the traditional pooling operations of simply selecting the maximum/average value within the window, BlurPool operations that use Gaussian-weighted averages can better preserve the relative positions and spatial relationships between features, thus alleviating the problem of information loss caused by pooling operations. BlurPool can be combined not only with pooling but also with strided-convolutions.

**Dilated convolution.** Traditional convolution operations only consider the point-wise product between the convolution kernel and the input. When using a dilated convolution, each specified position in the convolution kernel expands outward by a fixed number that is determined by the dilation rate. It enables the convolution kernel to process feature maps with a larger range of information while maintaining the same kernel size. In this way, by increasing the dilation rate, a larger receptive field can be obtained, and a wider range of image features can be captured. Dilated convolution captures more contextual information and improves the ability of models to handle high-resolution images. In traditional CNNs, as the resolution of the input image increases, the number of layers usually needs to increase accordingly. However, this leads to a larger model size and higher computational costs. By using dilated convolutions, a larger receptive field can be achieved without increasing the model size. Therefore, it can better process high-resolution images.

## 3.3 General Strategy

If we remove all downsampling operations, the dimension of image features will maintain vast, making the forward and backward computations too expensive to complete. Therefore, we adopt an "early stop" strategy for downsampling operations, which only retains downsampling in the early layers of the source model. In our experiments, we usually retain half of the downsampling operations in the original source model. This prevents losing too much low-level feature information and allows CNNs to consider the low-level feature information when generating adversarial samples. To further reduce the loss of feature information caused by downsampling, we modify the downsampling operations in the early layers of the source model with BlurPool as follows:

**MaxPool** $\rightarrow$ **MaxBlurPool**: For max-pooling with a stride of $s$, it is changed into two parts: max-pooling with a stride of 1 and BlurPool with a stride of $s$.

**Conv** $\rightarrow$ **ConvBlurPool**: For convolutions with a stride of $s$ followed by the ReLU activation function, it is changed into two steps: convolutions with a stride of 1 followed by the ReLU activation function and then BlurPool with a stride of $s$.

**AveragePool** $\rightarrow$ **BlurPool**: For average pooling with a stride of $s$, it is changed to BlurPool with a stride of $s$.

We then abandon the downsampling operations in the later layers of the source model. As a result of removing downsampling operations, the feature map size will increase, which reduces the receptive fields of subsequent layers and decreases the amount of context information each neuron has for making predictions [30]. Since contextual information is crucial for resolving ambiguities in local clues [31], this decrease in the receptive field is an undesired trade-off. Therefore, we use dilated convolutions to increase the receptive fields of higher layers, which compensates for the reduction in receptive fields caused by the removal of downsampling operations. This allows the modified source model to have the same or an even larger receptive field compared to the original model. In this way, the modified model can have a global understanding of the image, which enables it to generate adversarial perturbations that are more transferable.

Table 1: The structural details of Blurred-Dilated ResNet-50 (BD RN50).

| Layer Name | Output Size | Configuration |
|---|---|---|
| conv1 | 112×112 | 7×7 conv, 64, $s$=1
BlurPool, $s$=2 |
| pooling | 56×56 | 2×2 maxpool, $s$=1
BlurPool, $s$=2 |
| conv2
(×3) | 56×56 | 1×1 conv, 64
3×3 conv, 64
1×1 conv, 256 |
| conv3
(×4) | 28×28 | 1×1 conv, 128
3×3 conv, 128
(BlurPool when $s$=2)
1×1 conv, 512 |
| conv4
(×6) | 28×28 | 1×1 conv, 256
3×3 conv, 256, $dr$=2
1×1 conv, 1024 |
| conv5
(×3) | 28×28 | 1×1 conv, 512
3×3 conv, 512, $dr$=2
1×1 conv, 2048 |
| classification | 1×1 | global average pool |
| | 1000-d | FC-1000
softmax |

Table 2: The structural details of Blurred-Dilated DenseNet-121 (BD DN121).

| Layer Name | Output Size | Configuration |
|---|---|---|
| conv | 112×112 | 7×7 conv, $s$=1
BlurPool, $s$=2 |
| pooling | 56×56 | 3×3 maxpool, $s$=1
BlurPool, $s$=2 |
| dense layer1
(×6) | 56×56 | 1×1 conv
3×3 conv |
| transition layer1 | 56×56 | 1×1 conv |
| | 28×28 | BlurPool, $s$=2 |
| dense layer2
(×12) | 28×28 | 1×1 conv
3×3 conv |
| transition layer2 | 28×28 | 1×1 conv |
| dense layer3
(×24) | 28×28 | 1×1 conv
3×3 conv, $dr$=2 |
| transition layer3 | 28×28 | 1×1 conv |
| dense layer4
(×16) | 28×28 | 1×1 conv
3×3 conv, $dr$=2 |
| classification | 1×1 | global average pool |
| | 1000-d | FC-1000
softmax |

Finally, we fine-tune the modified BD model on its original training set to alleviate the accuracy drop caused by model modification. We employ the resultant BD models to generate adversarial samples to conduct transfer attacks.

### 3.4 BD Implementation Details

Our proposed BD can be conveniently applied to popular CNNs. Different source models may have slight differences in performing model modification. Therefore, we present the implementation details of BD with four popular architectures as examples in this section. In Tables 1-4, $s$ means the stride, $dr$ means the dilation rate, and BlurPool adopts a Gaussian kernel size of 4 and a stride of 2.

**Blurred-Dilated ResNet-50.** We apply our proposed BD to ResNet-50 [23] to obtain the Blurred-Dilated ResNet-50 (BD RN50). Table 1 shows the structural details of BD RN50. Specifically, we modify the first three downsampling operations with BlurPool. That is, for the first downsampling operation, we change the stride of the 7×7 convolution from 2 to 1, and then add a BlurPool layer. For the second downsampling operation, we change the stride of MaxPool from 2 to 1, and then add a BlurPool layer. For the third downsampling operation, we change the stride of conv3_1 from 2 to 1, and then add a BlurPool layer. At the same time, we remove downsampling operations in the subsequent layers to maintain the feature map size at 28×28. As shown in Table 1, dilated convolutions with a dilation rate of 2 are introduced in conv4 and conv5, respectively, except for conv4_1 and conv5_1.

**Blurred-Dilated DenseNet-121.** We apply our proposed BD to DenseNet-121 [27] to obtain the Blurred-Dilated DenseNet-121 (BD DN121). Table 2 shows the structural details of BD DN121. Specifically, we modify the first three downsampling operations with BlurPool. That is, we change the stride of the 7×7 convolution from 2 to 1, and then add a BlurPool layer. We change the stride of the 3×3 MaxPool from 2 to 1, and then add a BlurPool layer. We replace the first 2×2 average pool with a BlurPool layer. Furthermore, we remove the average pool in the last two transition layers. Similar to ResNet-50, dilated convolutions with a dilation rate of 2 are added to the final two groups of dense blocks as shown in Table 2, except for dense layer3_1 and dense layer4_1.

**Blurred-Dilated VGG16.** We apply our proposed BD to VGG16 [26] to obtain the Blurred-Dilated VGG16 (BD VGG16). Table 3 shows the structural details of BD VGG16. Specifically, the first three

Table 3: The structural details of Blurred-Dilated VGG16 (BD VGG16).

| Layer Name | Output Size | Configuration |
|---|---|---|
| conv0 (×2) | 224×224 | 3×3 conv, 64 |
| pooling1 | 112×112 | 2×2 maxpool, $s$=1 
 BlurPool, $s$=2 |
| conv1 (×2) | 112×112 | 3×3 conv, 128 |
| pooling2 | 56×56 | 2×2 maxpool, $s$=1 
 BlurPool, $s$=2 |
| conv2 (×3) | 56×56 | 3×3 conv, 256 |
| pooling3 | 28×28 | 2×2 maxpool, $s$=1 
 BlurPool, $s$=2 |
| conv3 (×3) | 28×28 | 3×3 conv, 512, $dr$=2 |
| conv4 (×3) | 28×28 | 3×3 conv, 512, $dr$=2 |
| classification | 7×7 | global average pool |
| | 1000-d | FC-4096 
 FC-4096 
 FC-1000 
 softmax |

Table 4: The structural details of Blurred-Dilated MobileNetV2 (BD MobV2). $t$ means the expansion factor. $c$ means the channel number.

| Layer Name | Output Size | Configuration | $t$ | $c$ |
|---|---|---|---|---|
| conv1 | 112×112 | 3×3 conv, $s$=1 
 BlurPool, $s$=2 | - | 32 |
| bottleneck1 (×1) | 112×112 | 1×1 conv 
 3×3 dwise 
 1×1 conv | 1 | 16 |
| bottleneck2 (×2) | 56×56 | 1×1 conv 
 3×3 dwise 
 (BlurPool when $s$=2) 
 1×1 conv | 6 | 24 |
| bottleneck3 (×3) | 28×28 | 1×1 conv 
 3×3 dwise 
 (BlurPool when $s$=2) 
 1×1 conv | 6 | 32 |
| bottleneck4 (×4) | 28×28 | 1×1 conv 
 3×3 dwise 
 1×1 conv | 6 | 64 |
| bottleneck5 (×3) | 28×28 | 1×1 conv 
 3×3 dwise, $dr$=2 
 1×1 conv | 6 | 96 |
| bottleneck6 (×3) | 28×28 | 1×1 conv 
 3×3 dwise, $dr$=2 
 1×1 conv | 6 | 160 |
| bottleneck7 (×1) | 28×28 | 1×1 conv 
 3×3 dwise 
 1×1 conv | 6 | 320 |
| conv2 | 28×28 | 1×1 conv | - | 1280 |
| classification | 1×1 | global average pool | - | - |
| | 1000-d | FC-1000 
 softmax | - | - |

MaxPool layers are replaced with MaxBlurPool. That is, the stride of the MaxPool layers is changed to 1, followed by a BlurPool layer with a Gaussian kernel size of 4 and a stride of 2. Downsampling layers in the later stages are also removed to retain the feature map's size at 28×28. Finally, as shown in Table 3, the last two groups of standard convolutions layers are replaced with dilated convolutions having a dilation rate of 2, except for conv3_1.

**Blurred-Dilated MobileNetV2.** We apply our proposed BD to MobileNetV2 [28] to obtain the Blurred-Dilated MobileNetV2 (BD MobV2). Table 4 shows the structural details of BD MobV2. Specifically, we stop downsampling when the feature map size reaches 28×28. To this end, in the bottlenecks of the original model that execute downsampling operations, we change the stride from 2 to 1. Moreover, for the first three convolutional downsampling operations, we change the stride of the convolution to 1 followed by a BlurPool layer. As shown in Table 4, dilated convolutions with a dilation rate of 2 are introduced in bottleneck5 and bottleneck6, except for bottleneck5_1 and bottleneck6_1.

## 4   Experiments

Section 4.1 details our experimental settings. In our experiments, we first verify the effectiveness of our proposed method in Section 4.2. Then in Section 4.3, we compare our proposed method with the state-of-the-art baselines. After that, we combine our proposed BD with other transfer attacks to further validate the effectiveness of our proposed method in improving adversarial transferability in Section 4.4. Finally, we conduct ablation studies to further analyze our proposed method in Section 4.5.

Table 5: The attack success rates of adversarial samples generated by different source models, with MI-FGSM as the optimization algorithm. The best results are in bold.

| Source Model | $\epsilon$ | RN50 | IncV3 | IncV4 | IncRes | SE154 | VGG19 | DN201 | PNASNet | Average |
|---|---|---|---|---|---|---|---|---|---|---|
| RN50 | 16/255 | **100%** | 67.0% | 57.1% | 54.5% | 69.2% | 82.8% | 88.3% | 63.7% | 72.8% |
| | 8/255 | **100%** | 44.9% | 37.1% | 30.3% | 47.6% | 63.3% | 73.3% | 39.3% | 54.5% |
| | 4/255 | **100%** | 22.9% | 19.0% | 10.6% | 25.5% | 43.8% | 47.4% | 17.9% | 35.9% |
| BD RN50 | 16/255 | 99.9% | **93.6%** | **93.8%** | **91.7%** | **95.1%** | **97.2%** | **99.1%** | **96.4%** | **95.9%** |
| | 8/255 | 99.8% | **80.1%** | **79.7%** | **74.6%** | **83.7%** | **91.2%** | **95.0%** | **81.0%** | **85.6%** |
| | 4/255 | 99.5% | **49.8%** | **47.7%** | **41.8%** | **58.1%** | **71.7%** | **75.9%** | **52.9%** | **62.2%** |
| VGG16 | 16/255 | 79.6% | 67.0% | 69.7% | 55.6% | 69.5% | 99.5% | 70.0% | 71.4% | 72.8% |
| | 8/255 | 58.9% | 41.0% | 43.2% | 28.0% | 44.9% | 98.1% | 46.0% | 44.0% | 50.5% |
| | 4/255 | 35.5% | 21.6% | 21.7% | 11.4% | 23.9% | 92.0% | 27.2% | 22.2% | 31.9% |
| BD VGG16 | 16/255 | **92.0%** | **82.0%** | **84.1%** | **73.5%** | **86.3%** | 99.8% | **87.3%** | **86.3%** | **86.4%** |
| | 8/255 | **78.6%** | **58.8%** | **63.5%** | **46.0%** | **64.7%** | 99.0% | **65.5%** | **66.4%** | **67.8%** |
| | 4/255 | **51.3%** | **31.6%** | **34.9%** | **18.9%** | **34.7%** | 95.7% | **39.0%** | **33.1%** | **42.4%** |
| DN121 | 16/255 | 90.3% | 71.7% | 63.8% | 60.2% | 73.3% | 85.0% | 95.3% | 69.7% | 76.2% |
| | 8/255 | 77.0% | 45.3% | 42.9% | 33.5% | 49.7% | 64.0% | 84.2% | 42.9% | 54.9% |
| | 4/255 | 52.6% | 26.3% | 21.6% | 12.1% | 25.7% | 43.4% | 62.3% | 20.2% | 33.0% |
| BD DN121 | 16/255 | **96.4%** | **84.5%** | **82.0%** | **76.4%** | **87.5%** | **94.3%** | 97.3% | **87.7%** | **88.3%** |
| | 8/255 | **86.7%** | **63.4%** | **59.0%** | **50.1%** | **66.6%** | **80.6%** | 89.5% | **65.1%** | **70.1%** |
| | 4/255 | **64.3%** | **36.3%** | **35.3%** | **21.6%** | **37.2%** | **59.5%** | 68.8% | **35.6%** | **44.8%** |
| MobV2 | 16/255 | 76.6% | 58.3% | 49.9% | 44.0% | 59.2% | 84.4% | 68.4% | 49.1% | 61.2% |
| | 8/255 | 54.4% | 36.1% | 30.4% | 20.8% | 38.0% | 63.7% | 47.6% | 26.2% | 39.7% |
| | 4/255 | 33.8% | 21.5% | 17.1% | 7.7% | 18.3% | 43.6% | 27.1% | 14.1% | 22.9% |
| BD MobV2 | 16/255 | **89.8%** | **82.7%** | **76.8%** | **69.5%** | **79.4%** | **95.0%** | **87.9%** | **76.9%** | **82.3%** |
| | 8/255 | **70.7%** | **60.1%** | **55.8%** | **44.6%** | **53.7%** | **81.7%** | **67.1%** | **51.5%** | **60.7%** |
| | 4/255 | **42.4%** | **32.6%** | **30.0%** | **18.2%** | **27.6%** | **59.5%** | **39.5%** | **24.6%** | **34.3%** |

## 4.1 Experimental Settings

**Datasets.** Consistent with the previous works [18, 21], we use the ImageNet-compatible dataset in the NIPS 2017 adversarial competition [32] as the test set to generate adversarial samples. This dataset contains 1000 images with a resolution of $299 \times 299 \times 3$. After modifying source models with our BD, we fine-tune the modified source models with the ImageNet training set [25] to recover their classification accuracy.

**Source and target models.** For source models, we choose one model from each of the ResNet [23], VGG [26], DenseNet [27], and MobileNet [28] families, i.e., ResNet-50 (RN50), VGG16, DenseNet-121 (DN121), and MobileNetV2 (MobV2). For comparisons with the state-of-the-art techniques, we follow their choice to use RN50 as the source model. For target models, we select popular models following previous works [13, 14, 15], including VGG19 [26], Inception-v3 (IncV3) [33], Inception-v4 (IncV4) [34], SENet154 (SE154) [35], ResNet-50 (RN50) [23], DenseNet-201 (DN201) [27], Inception-ResNet-v2 (IncRes) [34], and PNASNet [36]. We use open-source pretrained models that are collected from torchvision[1] and a GitHub repository[2].

**Compared baselines.** To validate the superiority of our proposed method, we compare it with the state-of-the-art model modification-based transfer attacks, including LinBP [15], ILA [12], and ILA++ [13].

**Parameter settings.** Whenever the size of the images does not match the model's input size, we resize the images to the input size of the model. We mainly focus on untargeted attacks under the $l_\infty$ constraint. We set the maximum perturbation $\epsilon$ to three different values: 16/255, 8/255, and 4/255.

---

[1]https://github.com/pytorch/vision/tree/main/torchvision/models
[2]https://github.com/Cadene/pretrained-models.pytorch

Table 6: The attack success rates of different transfer attacks, with MI-FGSM as the optimization algorithm and ResNet-50 as the source model. "Average" presents the average attack success rates against black-box models, i.e., all target models except ResNet-50. The best results are in bold.

| Method | $\epsilon$ | RN50 | IncV3 | IncV4 | IncRes | SE154 | VGG19 | DN201 | PNASNet | Average |
|--------|-----------|------|-------|-------|--------|-------|-------|-------|---------|---------|
| ILA | 16/255 | 99.9% | 73.8% | 69.3% | 68.1% | 80.8% | 91.0% | 92.0% | 73.6% | 78.4% |
| | 8/255 | **100%** | 47.8% | 43.0% | 38.2% | 53.7% | 74.5% | 72.7% | 42.5% | 53.2% |
| | 4/255 | **100%** | 24.5% | 21.7% | 13.7% | 29.5% | 49.6% | 49.8% | 20.4% | 29.9% |
| ILA++ | 16/255 | **100%** | 76.3% | 75.5% | 71.1% | 82.8% | 94.3% | 94.1% | 79.7% | 82.0% |
| | 8/255 | **100%** | 51.6% | 51.5% | 44.1% | 61.2% | 79.4% | 82.1% | 54.9% | 60.7% |
| | 4/255 | **100%** | 28.1% | 26.2% | 18.4% | 36.0% | 57.5% | 59.0% | 25.7% | 35.8% |
| LinBP | 16/255 | **100%** | 81.4% | 78.7% | 73.3% | 85.7% | 94.8% | 96.9% | 78.9% | 84.2% |
| | 8/255 | **100%** | 54.4% | 52.1% | 43.8% | 60.8% | 81.3% | 81.2% | 49.3% | 60.4% |
| | 4/255 | 99.9% | 30.0% | 25.1% | 16.1% | 33.0% | 59.4% | 54.9% | 21.5% | 34.3% |
| BD (Ours) | 16/255 | 99.9% | **93.6%** | **93.8%** | **91.7%** | **95.1%** | **97.2%** | **99.1%** | **96.4%** | **95.3%** |
| | 8/255 | 99.8% | **80.1%** | **79.7%** | **74.6%** | **83.7%** | **91.2%** | **95.0%** | **81.0%** | **83.6%** |
| | 4/255 | 99.5% | **49.8%** | **47.7%** | **41.8%** | **58.1%** | **71.7%** | **75.9%** | **52.9%** | **56.8%** |

Table 7: The attack success rates of different combined methods, with MI-FGSM as the optimization algorithm and ResNet-50 as the source model. "Average" presents the average attack success rates against black-box models, i.e., all target models except ResNet-50. The maximum perturbation $\epsilon = 16/255$. The best results are in bold.

| Method | RN50 | IncV3 | IncV4 | IncRes | SE154 | VGG19 | DN201 | PNASNet | Average |
|--------|------|-------|-------|--------|-------|-------|-------|---------|---------|
| ILA | 99.9% | 73.8% | 69.3% | 68.1% | 80.8% | 91.0% | 92.0% | 73.6% | 78.4% |
| BD+ILA | 99.9% | 95.0% | 94.2% | 94.2% | 96.6% | 99.1% | 99.1% | 95.6% | 96.3% |
| LinBP | **100%** | 81.4% | 78.7% | 73.3% | 85.7% | 94.8% | 96.9% | 78.9% | 84.2% |
| BD+LinBP | **100%** | 96.6% | 95.9% | 95.2% | 97.7% | **99.4%** | 99.4% | 96.7% | 97.3% |
| ILA+LinBP | **100%** | 85.2% | 84.9% | 80.5% | 89.0% | 97.7% | 97.7% | 85.8% | 88.7% |
| BD+ILA+LinBP | **100%** | **97.6%** | **97.2%** | **96.0%** | **97.9%** | **99.4%** | **99.5%** | **97.5%** | **97.9%** |

The step size $\alpha$ is set to 2/255 when $\epsilon = 16/255$, and 1/255 when $\epsilon = 8/255$ or 4/255. For MI-FGSM, we set the decay factor $\mu = 1.0$. The iteration number $T = 10$. ILA/ILA++ runs 10 iterations after 10 iterations' baseline attacks. For competitors, we adopt their official implementations. The parameter settings for the combined attacks are the same. All experiments were performed with an NVIDIA V100 GPU.

## 4.2 The Effectiveness of BD

Table 5 reports the attack performance of our method against different models. As we can see, when the source model is modified with our BD, the transferability of the generated adversarial samples is greatly improved across all source models. Specifically, our method improves the average attack success rates across all target models by 10.5%–31.1%. Notably, when the maximum perturbation is 16, the lowest adversarial transferability from ResNet-50 to target models is 54.5%. Our proposed BD increases it to 91.7%, which is an increase of 37.2%. The experimental results confirm the effectiveness of our proposed BD in generating highly transferable adversarial samples with different source models.

## 4.3 Comparisons with the Baselines

Table 6 reports the attack success rate of our BD and all the competitors against the target models, with MI-FGSM as the optimization algorithm and ResNet-50 as the source model. It can be seen that our proposed BD significantly outperforms the state-of-the-art model modification-based transfer attacks across all target models at different levels of perturbation budgets. Specifically, in the black-box scenarios, we achieve an average attack success rate of 95.3%, 83.6%, and 56.8%, under $\epsilon = 16/255$, 8/255, and 4/255, respectively. In contrast, the state-of-the-art baselines have the highest average

Table 8: The attack success rates of different BD variants, with MI-FGSM as the optimization algorithm and ResNet-50 as the source model. "Average" presents the average attack success rates against black-box models, i.e., all target models except ResNet-50. The maximum perturbation $\epsilon =$ 16/255. The best results are in bold.

| BD Variants | RN50 | IncV3 | IncV4 | IncRes | SE154 | VGG19 | DN201 | PNASNet | Average |
|---|---|---|---|---|---|---|---|---|---|
| BD w/ BlurPool only | 99.6% | 80.6% | 78.3% | 76.0% | 87.8% | 93.9% | 94.3% | 87.1% | 85.4% |
| BD w/ dilated conv only | 99.8% | 66.4% | 58.6% | 56.4% | 72.9% | 83.8% | 84.7% | 66.3% | 69.9% |
| BD | **100%** | **87.9%** | **86.8%** | **84.2%** | **93.5%** | **95.3%** | **98.0%** | **91.5%** | **91.0%** |

Table 9: The attack success rates on the CIFAR-10 and CIFAR-100 datasets, with MI-FGSM as the optimization algorithm. The best results are in bold.

| Dataset | Source Model | $\epsilon$ | WRN | ResNeXt | VGG19 | DN-BC | Average |
|---|---|---|---|---|---|---|---|
| CIFAR-10 | RN20 | 16/255 | 95.1% | 96.0% | 96.1% | 93.5% | 95.2% |
| | | 8/255 | 79.1% | 83.3% | 78.4% | 76.5% | 79.3% |
| | | 4/255 | 48.7% | 54.0% | 43.3% | 46.3% | 48.1% |
| | BD RN20 | 16/255 | **97.3%** | **97.8%** | **97.6%** | **96.0%** | **97.2%** |
| | | 8/255 | **84.7%** | **88.0%** | **82.4%** | **82.1%** | **84.3%** |
| | | 4/255 | **54.7%** | **60.8%** | **47.0%** | **51.8%** | **53.6%** |
| CIFAR-100 | RN56 | 16/255 | 95.4% | 94.4% | 94.3% | 93.3% | 94.4% |
| | | 8/255 | 82.3% | 80.5% | 82.2% | 78.8% | 81.0% |
| | | 4/255 | 60.4% | 58.5% | 61.8% | 57.3% | 59.5% |
| | BD RN56 | 16/255 | **97.2%** | **96.8%** | **96.1%** | **96.0%** | **96.5%** |
| | | 8/255 | **87.1%** | **85.6%** | **85.0%** | **85.2%** | **85.7%** |
| | | 4/255 | **66.0%** | **64.5%** | **64.9%** | **64.1%** | **64.9%** |

attack success rates of only 84.2%, 60.7%, and 35.8% under the same settings, respectively. In the white-box setting, since BD ResNet-50 is different from the original ResNet-50, and we use BD ResNet-50 to generate adversarial samples, the attack success rates are slightly lower than 100%. However, we note that the performance drop is negligible.

## 4.4 Combining with Existing Methods

Since our BD preserves the core architecture of the source model, it can be combined with previous transfer attacks such as LinBP and ILA. The results are presented in Table 7, with MI-FGSM as the optimization algorithm and ResNet-50 as the source model. We can see that our BD can significantly further promote the transferability of adversarial samples when combined with other transfer attacks. Specifically, BD significantly improves the black-box attack success rates of ILA and LinBP by an average of 17.9% and 13.1%, respectively. When combining BD with ILA and LinBP in the black-box settings, the average attack success rate can reach 97.9%, and the lowest attack success rate can be increased to 96.0%, compared to only 80.5% achieved by combining ILA and LinBP.

## 4.5 Further Analysis

To verify the contribution of each component in BD, we conduct an ablation study by removing the BlurPool component and the dilated convolution component from BD, respectively. Table 8 reports the attack success rates, with MI-FGSM as the optimization algorithm and ResNet-50 as the source model. The maximum perturbation is set to 16/255. We can see that the removal of the BlurPool component and the removal of the dilated convolution component lead to a decrease in black-box attack performance by 21.1% and 5.6% on average, respectively. These results suggest that BlurPool and dilated convolutions both contribute to improving adversarial transferability, which validate the design of our method.

Besides, we perform experiments on datasets with lower resolution. Following previous efforts [15, 13], we choose the CIFAR-10 and CIFAR-100 datasets [24], which consist of 60000 images

from 10 and 100 classes, respectively. They are both officially divided into a training set of 50000 images and a test set of 10000 images. We use their test sets to generate adversarial samples and their training sets to fine-tune the modified BD model. We choose ResNet-20 (RN20) and ResNet-56 (RN56) [23] as the source models for the CIFAR-10 and CIFAR-100 datasets, respectively. For target models, we choose widely applied DNN models, including Wide Residual Network (the WRN-28-10 version) [37], ResNeXt-29 [38], VGG19 [26], and DenseNet-BC (DN-BC) [27]. We use open-source pretrained models that are collected from a GitHub repository[3].

Table 9 presents the results. We can see that our method can still consistently improve the adversarial transferability of the source model by a large margin, which confirms that our method is also effective on datasets with lower resolution.

## 5 Conclusion and Discussion

In this paper, we propose a novel model modification-based transfer attack: Blurred-Dilated method (BD). BD reduces downsampling operations and adds BlurPool and dilated convolutions in the original source model. It then employs the modified source model to generate adversarial samples. Extensive experiments on the ImageNet dataset validate that BD can be conveniently applied to different popular source models to generate highly transferable adversarial examples. Our proposed method can also obtain significantly better adversarial transferability than the state-of-the-art baselines. Besides, our method can be conveniently combined with other transfer attacks to further improve their performance.

**Limitations and future work.** Our proposed model modification method is mainly based on domain knowledge and empiricism. Therefore, although there is a general strategy, it still requires manual trial and error to attain the final model structures that can generate highly transferable adversarial samples. Moreover, the final model structure that we obtain is only the optimal one among the candidate structures that we consider in this work. We leave it to future work to automate the search for model structures that can manufacture highly transferable adversarial examples. Our work also mainly focuses on CNN models that have convolutional and pooling layers. We plan to extend our method to more non-CNN models, like Vision Transformers [39] in the future work.

**Broader impacts.** Our work not only offers a new way to effectively evaluate the robustness of deep learning models, but also provides further insights into the relationship between model structures and adversarial transferability. Additionally, our work calls the attention of both researchers and practitioners to this new security issue. We believe that our method can be utilized to generate more transferable adversarial samples for adversarial training, which can better improve the robustness of a model.

## Acknowledgments and Disclosure of Funding

The authors are grateful to the anonymous reviewers for their valuable comments and suggestions, which helped to improve this paper. This work was supported by the National Natural Science Foundation of China (Grant No. 62206318).

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
