# Blurred-Dilated Method for Adversarial Attacks (Supplementary Material)

**Yang Deng**
School of Software Engineering
Sun Yat-sen University
dengy73@mail2.sysu.edu.cn

**Weibin Wu**[*]
School of Software Engineering
Sun Yat-sen University
wuwb36@mail.sysu.edu.cn

**Jianping Zhang**
Department of Computer Science and Engineering
The Chinese University of Hong Kong
jpzhang@cse.cuhk.edu.hk

**Zibin Zheng**
School of Software Engineering
Sun Yat-sen University
zhzibin@mail.sysu.edu.cn

## A  The Structural Details of BD Models on CIFAR-10/100

Table S1: The structural details of Blurred-Dilated ResNet-20 (BD RN20).

| Layer Name | Output Size | Configuration |
|---|---|---|
| conv1 | 32×32 | 3×3 conv, 16, $s$=1 |
| conv2 (×3) | 32×32 | 3×3 conv, 16
3×3 conv, 16 |
| conv3 (×3) | 16×16 | 3×3 conv, 32
(BlurPool when $s$=2)
3×3 conv, 32 |
| conv4 (×3) | 16×16 | 3×3 conv, 64, $dr$=2
3×3 conv, 64 |
| classification | 1×1 | global average pool |
|  | 10-d | FC-10
softmax |

Table S2: The structural details of Blurred-Dilated ResNet-56 (BD RN56).

| Layer Name | Output Size | Configuration |
|---|---|---|
| conv1 | 32×32 | 3×3 conv, 16, $s$=1 |
| conv2 (×6) | 32×32 | 1×1 conv, 16
3×3 conv, 16
1×1 conv, 64 |
| conv3 (×6) | 16×16 | 1×1 conv, 32
3×3 conv, 32
(BlurPool when $s$=2)
1×1 conv, 128 |
| conv4 (×6) | 16×16 | 1×1 conv, 64
3×3 conv, 64, $dr$=2
1×1 conv, 256 |
| classification | 1×1 | global average pool |
|  | 100-d | FC-100
softmax |

Table S1 presents the structural details of Blurred-Dilated ResNet-20 (BD RN20) used on the CIFAR-10 dataset. Table S2 shows the structural details of Blurred-Dilated ResNet-56 (BD RN56) used on the CIFAR-100 dataset. In both tables, $s$ means the stride, $dr$ means the dilation rate, and BlurPool adopts a Gaussian kernel size of 4 and a stride of 2.

## B  More Ablation Studies

We conduct more ablation studies to examine the key modification choices of our method. Specifically, we examine: (1) which downsampling layer should be modified with BlurPool, and (2) the dilation rates of the last two sets of dilated convolutions. The source and target models are the same as those

---

[*]Corresponding author.

37th Conference on Neural Information Processing Systems (NeurIPS 2023).

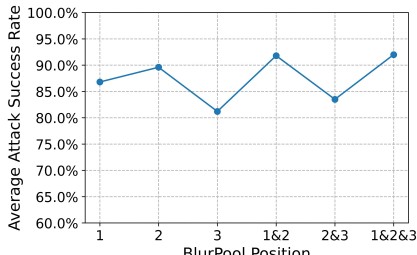 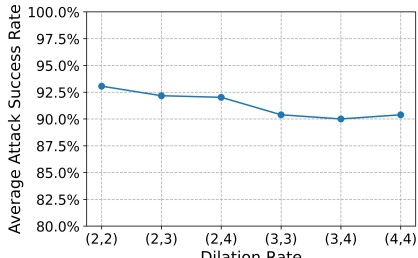

Figure S1: The average attack success rates against all the target models under different modification configurations, with ResNet-50 as the original source model. BlurPool position $i$ means that the BlurPool is applied to the $i$-th downsampling operation. Dilation rate $(i,j)$ means that the dilation rates for the first and second groups of dilated convolutions are $i$ and $j$, respectively.

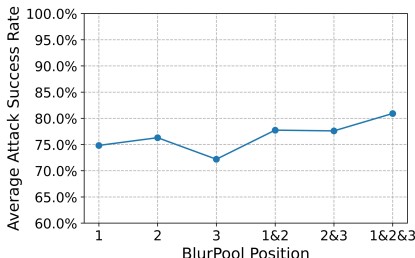 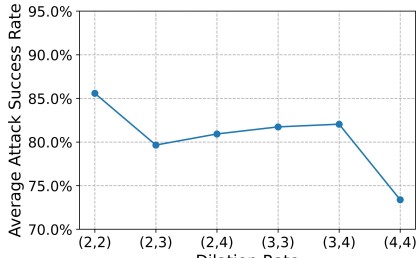

Figure S2: The average attack success rates against all the target models under different modification configurations, with DenseNet-121 as the original source model. Notations are the same as Figure S1.

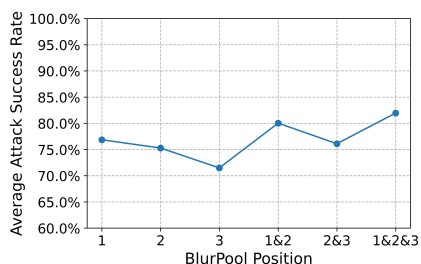 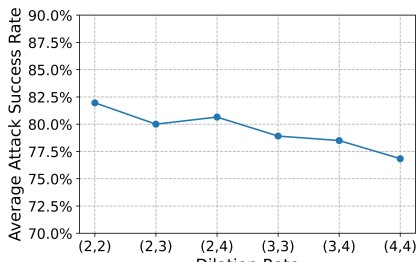

Figure S3: The average attack success rates against all the target models under different modification configurations, with VGG16 as the original source model. Notations are the same as Figure S1.

in the main paper. Based on our preliminary experiments, we focus on testing six candidate positions where BlurPool is added: (1) the first downsampling operation, (2) the second downsampling operation, (3) the third downsampling operation, (4) the first and second downsampling operations, (5) the second and third downsampling operations, and (6) all the first three downsampling operations. Similarly, for the dilation rates of the last two sets of dilated convolutions, we focus on testing six possible combinations: (2,2), (2,3), (2,4), (3,3), (3,4), and (4,4).

Figures S1-S4 show the average attack success rates against all the target models under different modification configurations. It is evident that across different source models, our method achieves the best results under the same modification configuration, i.e., when BlurPool is combined with all the first three downsampling operations, and the dilation rates are (2,2). Therefore, our modification configuration can be a good start point when modifying other source models. We note that since BD models reduce downsampling operations during forward propagation, the inference time of a BD

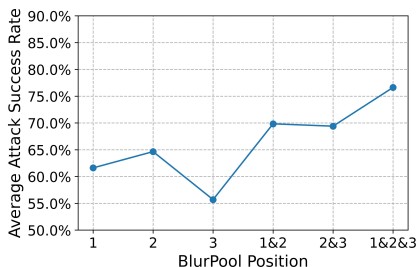 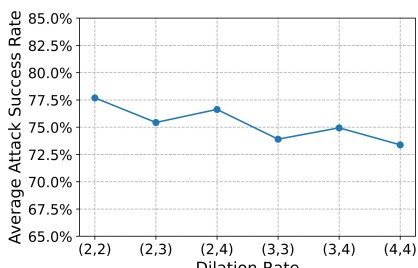

Figure S4: The average attack success rates against all the target models under different modification configurations, with MobileNetV2 as the original source model. Notations are the same as Figure S1.

Table S3: The attack success rates of different transfer attacks against adversarially trained models, with MI-FGSM as the optimization algorithm and ResNet-50 as the source model. The best results are in bold.

| Method | $\epsilon$ | IncV3$_{ens3}$ | IncV3$_{ens4}$ | IncRes-v2$_{ens}$ | Average |
|---|---|---|---|---|---|
| ILA | 16/255 | 48.5% | 44.1% | 37.3% | 43.3% |
| | 8/255 | 26.6% | 26.5% | 16.6% | 23.2% |
| | 4/255 | 14.8% | 16.6% | 8.3% | 13.2% |
| ILA++ | 16/255 | 53.0% | 47.2% | 38.3% | 46.2% |
| | 8/255 | 31.3% | 30.8% | 19.9% | 27.3% |
| | 4/255 | 16.2% | 18.6% | 9.3% | 14.7% |
| LinBP | 16/255 | 59.9% | 55.6% | 46.8% | 54.1% |
| | 8/255 | 34.3% | 33.1% | 22.4% | 29.9% |
| | 4/255 | 16.9% | 18.7% | 8.7% | 14.8% |
| BD (Ours) | 16/255 | **84.1%** | **80.6%** | **73.3%** | **79.3%** |
| | 8/255 | **63.6%** | **58.2%** | **47.6%** | **56.5%** |
| | 4/255 | **32.7%** | **30.8%** | **21.1%** | **28.2%** |

model is relatively longer than the original standard model. However, the attack success rates we can obtain are significantly higher than the original standard model.

## C The Effectiveness of BD against Defenses

We validate the effectiveness of our proposed BD against defenses. We first attack adversarially trained models, including IncV3$_{ens3}$, IncV3$_{ens4}$, and IncRes-v2$_{ens}$ [S1]. The results are shown in Table S3. Then we consider other advanced defenses, including JPEG [S2], FD [S3], FAT [S4], RS [S5], and NRP [S6]. The results are shown in Table S4. Under both settings, our attack still outperforms all the state-of-the-art baselines by a large margin. The results confirm the effectiveness of our method against defended models, which also calls for the development of stronger defenses.

## D Visualization

We investigate which features of the input image are emphasized by our BD models and standard models during the inference process. To this end, in Figure S5, we visualize the attention maps of the standard models and the modified BD models to examine the critical ground for their predictions. We can see that the attention region of the BD models aligns better with the object's important features. In contrast, the attention region of the standard models appears to cover a lot of unnecessary information. Therefore, BD models can more precisely extract the object's important features than the standard models. It seems to explain the effectiveness of our approach from another perspective. Due to the

Table S4: The attack success rates of different transfer attacks against other advanced defenses, with MI-FGSM as the optimization algorithm and ResNet-50 as the source model. The maximum perturbation is 16/255. The best results are in bold.

| Method | JPEG | FD | FAT | RS | NRP | Average |
|---|---|---|---|---|---|---|
| ILA++ | 56.8% | 45.6% | 34.8% | 28.6% | 17.8% | 36.7% |
| LinBP | 63.8% | 59.7% | 39.7% | 35.9% | 23.0% | 44.4% |
| BD (Ours) | **83.9%** | **77.5%** | **43.6%** | **54.7%** | **35.8%** | **59.1%** |

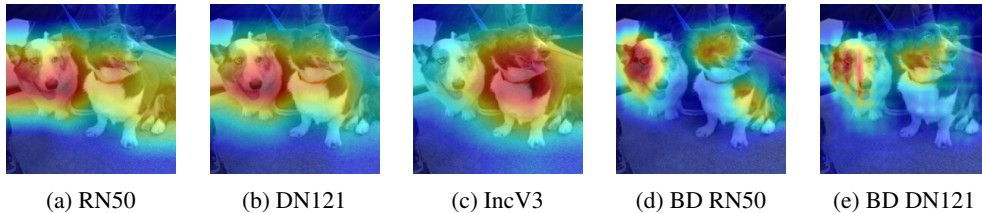

(a) RN50  (b) DN121  (c) IncV3  (d) BD RN50  (e) BD DN121

Figure S5: The attention maps of different models.

BD model's ability to more precisely extract important features of objects, when using the BD model as the source model, the generated adversarial sample will pay more attention to interfering with important features of objects, which are also used by different models for object classification. In contrast, the adversarial perturbation generated by the standard models may focus on features that are extraneous to the object classification, which may not be the focus of other models [S7]. Therefore, our BD models can generate more transferable adversarial samples than the standard models.