# OpenReview forum: "Blurred-Dilated Method for Adversarial Attacks"
_NeurIPS.cc/2023/Conference — NeurIPS 2023 poster_

### Official Review · Reviewer_eif6 · 2023-07-03

**Soundness:** 3 good
**Presentation:** 3 good
**Contribution:** 3 good
**Rating:** 6
**Confidence:** 4

**Summary:**

The authors propose the Blurred-Dilated method (BD), which utilizes BlurPools and dilated convolutions on the source model when an adversarial attack is applied, to increase the transferability of the transfer-based attack. The method replaces the MaxPool layer with MaxBlurPool, Conv with ConvBlurPool and AveragePool with BlurPool, in both forward and backward computation. The author conducts experiments and find that BD can outperform multiple SOTA. On top of that, Combining BD with existing black-box attacks can further improve the attack success rate.

**Strengths:**

1. The paper is written very clearly with a detailed introduction to the preliminary works. The actual modifications to the models are detailedly reported in tables.
2. A large variety of experiments are covered, such as comparison with SOTA, success rates against robustly trained models, ablation study, hyper-parameters, etc. A lot of potential concerns can be addressed with the reported results.
3. The proposed method is simple yet effective. Replacing model layers does not introduce extra computation time compared with methods like GhostNet.


**Weaknesses:**

#### 1. Datasets with lower resolution are not tested with
In the paper, all the experiments are performed on ImageNet. Since BD modifies the downsampling operation, which can make a huge difference with different image resolutions. ImageNet has a relatively high resolution (3x299x299), enabling dilated convolution to be applied without much issue. However, I wonder if BD can still generate transferable attacks for datasets with lower resolution, such as CIFAR-10 and CIFAR-100.

#### 2. The proposed method is specific to some model components
This limitation is also brought up by the authors, "*Our proposed model modification is based on domain knowledge and empiricism.*" Besides, there is no general guideline on the "early stop" strategy on how many downsampling layers to be removed. Summing up these points, it can be difficult to extend the proposed methods to new models.

For the same reason, claims like line 176 might be too broad. "*BD (Blurred-Dilated method) is a universal technique that can be easily implemented in any DNN.*" As BD requires specific CNN layers like Convolution with stride and pooling layers to work, it cannot be directly applied to non-CNN models like vision transformers and MLP mixers. This is also another limitation that is not discussed in the paper.

#### 3. Minor formatting recommendations
- I recommend inserting a space between the square bracket for citation and the previous word (e.g. word [1] instead of word[1])
- The formatting of the norm is inconsistent in the main text (l$\infty$) and the appendix ($l_\infty$).


**Questions:**

1. If we consider replacing max pooling with BlurPool, can we interpret it as replacing a non-linear function with a linear function? Thus, does the Linearity Hypothesis in LinBP apply to BD as well?

2. I suppose the replacement of model layers takes place during test time (actually it is better to clarify in the paper), when the source models are already pre-trained similar to SGM and LinBP. However, unlike SGM and LinBP, BD also modifies forward propagation. As the architecture is changed, I wonder if we can retrain/finetune the modified model and uses the new one as the source model instead?


**Limitations:**

The authors admit Weakness #2 in the paper. However, can the authors advise some guidelines on the 'early stop' strategy and what is the proportion of the low-level features to be discarded/retained?

---

> ### Author Rebuttal · Authors · 2023-08-10
>
> Q1. If we consider replacing max pooling with BlurPool, can we interpret it as replacing a non-linear function with a linear function? Thus, does the Linearity Hypothesis in LinBP apply to BD as well?
>
> Thank you for providing a new perspective to explain the effectiveness of our method. However, we actually replace max pooling with MaxBlurPool, which consists of max-pooling with a stride of 1 and blur-pooling (Line 161 of our main paper). Therefore, MaxBlurPool is still a non-linear function.
>
> Furthermore, LinBP proposes that the linear structure in the network is helpful for attack transferability, and removes the ReLU function in the network. Maybe it's because the ReLU function causes information loss when x<0?
>
> Q2. Can we retrain/finetune the modified model and uses the new one as the source model instead?
>
> Sorry, we didn't explain this clearly. In fact, we have fine-tuned the model after modifying it with the BD method. Otherwise the accuracy of the modified model will decrease, degrading the attack performance as well. We will make it more clear in our final version.
>
>
> Q3. Experiments on lower-resolution datasets.
>
> We experiment with lower-resolution datasets: CIFAR-10 and CIFAR-100. The experimental results are shown in Table R1 of the uploaded PDF file. The original source models for CIFAR-10 and CIFAR-100 are ResNet 20 and ResNet 56, respectively. We can see that our method can still be applied to lower-resolution datasets to generate transferable adversarial samples. Besides, we still consistently outperform the baseline method by about 4\% on average.
>
> Q4. Can the authors advise some guidelines on the 'early stop' strategy and what is the proportion of the low-level features to be discarded/retained?
>
> (1) Through experiments, we found that retaining half of the downsampling operations in the model can be a guideline.
>
> (2) The input size of ResNet is $224\times224$, and a $7\times7$ feature map is obtained finally after 5 downsampling operations. Therefore, the proportion of the low-level features to be discarded/retained for the original ResNet is $(224^2-7^2)/7^2$. In contrast, in our BD ResNet, we only keep 2 or 3 downsampling operations, obtaining a $4^2$ times larger feature map ($28\times28$) finally. Therefore, the proportion of the low-level features to be discarded/retained is $(224^2-28^2)/28^2$.
>
>
> Q5. Fixing the claims like line 176.
>
> Our claim is not accurate, we will change it to "BD (Blurred-Dilated method) is ageneral technique that can be easily implemented in popular CNNs."
>
> Q6. Minor formatting recommendations.
>
> Thanks for your careful review. We will fix this typo and thoroughly proofread the paper again.

---

> > ### Comment · Reviewer_eif6 · 2023-08-14
> > **Thanks for the Rebuttal**
> >
> > Weakness #1: The experiments in Table R1 resolve my concerns. The authors successfully show that their method (BD) can also work well on datasets with lower resolution such as CIFAR-10 and CIFAR-100.
> >
> > Weakness #2: This is undeniably the limitation of the proposed method. Nevertheless, in my opinion, this alone is insufficient to lead to a "reject". Some of the broad claims need to be fixed, which is acknowledged and promised by the authors.
> >
> > Questions: Q2, 5, 6 show that there is room for improvement in the paper in clarity and formatting. The authors also promised to improve the clarity in these specific areas.
> >
> > For the reasons above, I will keep my rating. However, I would like to point out that the design choice of the number of features to be discarded/retained seems to be obtained mainly from experiments. The motivation/justification can be made stronger if the authors also include a more detailed discussion similar to the rebuttal to reviewer hwet and their new findings from CIFAR-10 and CIFAR-100 in the paper.

---

> > > ### Author Response · Authors · 2023-08-18
> > > **Thank you for the comment!**
> > >
> > > We are glad to hear that our response addresses your concerns. We sincerely appreciate the positive comments on the impact and contribution of this work. We will revise our manuscript to correct the formatting issues and make it more clear. We will also include a more detailed discussion to make our motivation and justification stronger based on your suggestions. We are grateful for your careful consideration of this work and insightful comments!

---

### Official Review · Reviewer_MuF1 · 2023-07-04

**Soundness:** 3 good
**Presentation:** 2 fair
**Contribution:** 2 fair
**Rating:** 5
**Confidence:** 3

**Summary:**

In this work, the author generates adversarial examples to attack other models by using BlurPools and dilated convolutions on the source model. The results show that increasing the model with BlurPools and dilated convolutions can generate more transferable adversarial examples.

**Strengths:**

The work is well-written and easy to follow. The work performs comprehensive experiments and validates the effectiveness of BlurPools and dilated convolutions regarding improving the transferability of adversarial examples.

**Weaknesses:**

Although the author has demonstrated the effectiveness of the proposed method through experiments, my main point is still that this work lacks sufficient novelty. In my opinion, adding a blur layer inside the network and methods based on input augmentation (such as padding and resizing) are not fundamentally different. Moreover, the former method actually has more model dependence and cannot be used as plug-and-play as the latter.

Additionally, some other concerns include:
1. "They mostly focus on backpropagation while neglecting forward propagation." This limitation is not entirely correct. According to the author's classification, input augmentation can also be considered an improvement from forward propagation. I believe this statement should be further explained in more detail.

2. "It is crucial to retain as many comprehensive features as possible." This statement may not be entirely rigorous. The best transferability comes from category-related features, and background features sometimes provide false information.

3. In terms of experimental results, there is still some gap compared to some existing sota methods.

**Questions:**

See weakness.

**Limitations:**

See weakness.

---

> ### Author Rebuttal · Authors · 2023-08-10
>
> Q1. The novelty of the proposed method.
>
> Please see Q2 of Reviewer hwet.
>
> Q2.  "In my opinion, adding a blur layer inside the network and methods based on input augmentation (such as padding and resizing) are not fundamentally different."
>
> (1) Our method can help to preserve more low-level features during forward propagation of the modified source model. Therefore, employing such a source model can guide adversarial samples to more thoroughly and precisely destroy the low-level features of a clean image. Such adversarial samples are more transferable, since different models all use low-level features to extract high-level semantics and then make predictions (Please see Q1 of Reviewer hwet).
>
> (2) In contrast, input augmentation improves adversarial transferability by offering less noisy gradients to escape from the poor local optimal of a source model. Therefore, our method and attacks based on input augmentation are fundamentally different. Besides, we can combine our method with input augmentation based attacks to further improve the performance.
>
>
> Q3. Explaining "They mostly focus on backpropagation while neglecting forward propagation."
>
> We are sorry about that. "They" here refer to the attack methods based on model modification. Many existing transfer attacks based on model modification mainly focus on modifying the back propagation.
>
>
> Q4. Fixing "It is crucial to retain as many comprehensive features as possible.''
>
> We will change it to "It is crucial to retain as much low-level features as possible.'' Please see Q1 of Reviewer hwet for explanations.
>
> Q5. Comparing with existing SOTA methods in terms of the experimental results.
>
> (1) Our work is focused on model-modification based attacks. LinBP and ILA family are SOTA methods in this category, and we have compared with them.
>
> (2) Perhaps the SOTA method you mentioned is based on other mechanisms such as input augmentation. To address your concerns, we compared our method with the state-of-the-art input-augmentation based method SSA [R6] with a maximum perturbation of 16, using MI as the attack method. Table R2 of the uploaded PDF file shows the attack results. We can see that our method can achieve an average attack success rate of 95.3\%, outperforming SSA by 1.6\%. Moreover, our method can be combined with SSA to further improve the attack success rates of SSA by 4.7\% on average.
>
>
> [R6] Long, Y., Zhang, Q., Zeng, B., Gao, L., Liu, X., Zhang, J., & Song, J. (2022, October). Frequency domain model augmentation for adversarial attack. In European Conference on Computer Vision (pp. 549-566). Cham: Springer Nature Switzerland.

---

> ### Author Response · Authors · 2023-08-18
> **Follow-up response**
>
> Dear Reviewer,
>
> Considering that the discussion phase is nearing to end, we are looking forward to your further feedback about our latest response. Do our responses fully address your concerns? Do you have any other comments? We would like to discuss with you in more detail. We greatly appreciate your time and feedback.
>
> Sincerely,
>
> Authors

---

> > ### Comment · Reviewer_MuF1 · 2023-08-20
> >
> > Thanks a lot for your response. Most of my concerns have been addressed. However, I still have a slight question regarding the comparison with state-of-the-art (SOTA) attacks. I believe that the ultimate goal of this work should be to achieve state-of-the-art transferability results, considering the motivation the concept of __low-level feature improving transferability__ has been mentioned to some extent in other works (ILA etc). If we follow the authors' statement and compare it with model-modification based attacks like [1], what would be the advantages of this work?
> >
> > [1] Towards Understanding and Boosting Adversarial Transferability from a Distribution Perspective.

---

> > > ### Author Response · Authors · 2023-08-21
> > >
> > > Q1. Comparing with DRA [1].
> > >
> > > (1) Due to the limited time left, we can only conduct some preliminary experiments to compare our method with DRA [1], when using ResNet-50 as the source model and MI-FGSM as the attack algorithm. The attack success rates of both methods on ImageNet are shown below. We can see that our method can still outperform DRA [1].
> > >
> > >
> > > |  ε |  Target Model   |   DRA    |   BD (Ours)   |
> > > |:-----:|:---------:|:-----:|:-----:|
> > > |  16  |  RN50 | 99.6%  |  **99.9%** |
> > > |  16  |  PNAS | 93.8%  |  **96.4%** |
> > > | 8    | RN50 |  95.9% |  **99.8%** |
> > > |  8   | PNAS | 63.8%  | **81.0%**  |
> > > |   4  | RN50 |  71.7% | **99.5%** |
> > > |  4   | PNAS |  23.6% | **52.9%** |
> > >
> > >
> > > (2) DRA does not modify the structure of the source model to make it able to preserve more low-level features of an image. Instead, DRA proposes to improve adversarial transferability from the data distribution perspective. It defines a new loss function to fine-tune a source model so that the gradient of the fine-tuned source model can approximate the gradient of the ground-truth data distribution. As a result, using the fine-tuned source model can better push the image away from its original distribution, which helps to improve adversarial transferability.
> > >
> > > (3) We can combine our method with DRA [1] to achieve better performance, since these two methods attempt to improve adversarial transferability from different perspectives.
> > >
> > >
> > > Q2. "the ultimate goal of this work should be to achieve state-of-the-art transferability results, considering the motivation the concept of low-level feature improving transferability has been mentioned to some extent in other works (ILA etc)."
> > >
> > > We do not agree that ILA has mentioned the concept of low-level features improving transferability. The motivation of ILA is to increase the perturbation of an adversarial sample on a pre-specified layer of the source model, which the authors hope will be conducive to greater transferability. Therefore, in addition to achieving state-of-the-art transferability results, our work contributes to providing a new perspective to study the transferability of adversarial samples.

---

> > > > ### Comment · Reviewer_MuF1 · 2023-08-22
> > > >
> > > > Thanks for your reply. I have raised my score accordingly.

---

### Official Review · Reviewer_1sbs · 2023-07-04

**Soundness:** 3 good
**Presentation:** 3 good
**Contribution:** 2 fair
**Rating:** 5
**Confidence:** 4

**Summary:**

This paper proposes a new Blurred-Dilated method for generating transfer attacks. The authors focus on generating transfer attacks as a more realistic attack model by looking at how the substitute model's architecture can be changed to increase the transferability of adversarial attacks. By introducing blurred downsampling and dilated convolutions in the substitute network, the authors try to focus on preserving important features to increase transferability. The authors evaluate transferability on ImageNet across several naturally trained architectures.


======= POST REBUTTAL =========
After the rebuttal, I have raised my score due to experiments on transfer-based defenses.

**Strengths:**

1. Interesting idea. The paper proposes an interesting approach, which is whether or not transfer attacks can be carried out on the substitute model side. If there are techniques we can do on the substitute model side, this makes the attacks more possible and practical.

2. Compared to ILA, ILA++, and LinBP, the BD attacks seem to do quite well, transferring across naturally trained architectures at a higher rate than these other attacks.

3. Examining the attention maps in the experiments is interesting, and helps add some insight into what would otherwise be an empirical paper.

**Weaknesses:**

1. Some design choices could be defended stronger. For example, why are the dilated convolutions only applied at the later layers? Why is blurred-downsampling better than convolving with higher stride? Also, the blurred filters would still lose information. What formally is being preserved with the BD filters and not with the regular convolutions or max pooling or average pooling? More formal one-to-one comparisons between each choice may be helpful. I am still a bit unsure about what fundamentally is important about these operations.

2. Unstated practicality concerns. While the results are good against naturally trained models, what if the defense is trained to stop transfer attacks? E.g., "Ensemble Adversarial Training: Attacks and Defenses" (Tramer et al. 2018) or "TRS: Transferability Reduced Ensemble via Encouraging Gradient Diversity and Model Smoothness" (Yang et al. 2021). The introduction of such defenses may render these attacks ineffective. I wonder how effective these defenses would be in stopping BD attacks. In addition, the assumption is that ImageNet as the domain is known but the architecture may be different. How realistic is this in the real world? One might not know the dataset that a machine learning model was trained on, and then would have to extract a substitute with model extraction. Then, the practical cost of extracting a model and then performing a transfer attack would have to be compared with the cost of performing query-based black-box attacks.

**Questions:**

1. Why are dilated convolutions only applied at the later layers?

2. Why are blurred-downsampling filters better than convolving with higher stride?

3. How does BD perform against defenses trained to stop transfer attacks?

**Limitations:**

The paper is an attack paper and does not extensively talk about the implications of such a method. A discussion on possible countermeasures and how this impacts the overall aim of creating robust and reliable machine learning models would be helpful.

---

> ### Author Rebuttal · Authors · 2023-08-10
>
> Q1. Why are dilated convolutions only applied at the later layers?
>
> (1) We introduce dilated convolutions to reduce downsampling in the model, since we want to preserve more low-level features of an image during forward propagation (Please see the answer to Q1 of Reviewer hwet for the motivation of our method). However, merely removing downsampling can degrade the model's performance on capturing the global information of an image, since downsampling can enlarge the model's receptive field. Therefore, in addition to removing downsampling, we add dilated convolutions to enlarge the model's receptive field while maintaining more low-level features.
>
> (2) We do not apply dilated convolutions to reduce downsampling at the earlier layers, since applying dilated convolutions at the earlier layers will maintain the large dimensionality of earlier features, which makes the forward computation too expensive to complete.
>
>
> Q2. Why are blurred-downsampling filters better than convolving with higher strides?
>
> In addition to adding blurred-downsampling, we change convolving with higher strides to convolving with stride 1. Such a combination (ConvBlurPool, please see Line 163 of the main paper) is better than convolving with higher strides in terms of preserving more low-level and low-frequency features. The reasons are:
>
> (1) Convolving with higher strides would skip pixels and potentially miss features in the original image. Therefore, we first change the stride to 1, which can preserve more low-level features.
>
> (2) We then add blurred-downsampling. Although it reduces the dimensionality of features, the output of blurred-downsampling still considers all features in its receptive field. Therefore, it will not cause more information loss than convolving with higher strides.  Besides, since blurred-downsampling is a Gaussian filter, it can keep more low-frequency features, which can help to generate more transferable adversarial samples [6].
>
>
>
> Q3. How does BD perform against defenses trained to stop transfer attacks?
>
> We have tested three adversarially trained models: IncV3\textsubscript{ens3}, IncV3\textsubscript{ens4}, and IncRes\textsubscript{ens} (Tramer et al. 2018). The detailed results are shown in Table 9 of our appendix due to space limit. TRS (Yang et al. 2021) does not provide a pre-trained defense model, and we cannot complete the model training for TRS due to the limited rebuttal period. Therefore, we tested more widely-used defenses to stop transfer attacks: JPEG [R1], FD [R2], FAT [R3], RS [R4] and NRP [R5]. The results are shown in Table R3 of the uploaded PDF file. We can see that these defenses cannot effectively stop our BD attacks, and we outperform the state-of-the-art baselines by a large margin of 14.7\% on average.
>
>
> [R1] Guo, C., Rana, M., Cisse, M., & Van Der Maaten, L. (2017). Countering adversarial images using input transformations. arXiv preprint arXiv:1711.00117.
>
> [R2] Liu, Z., Liu, Q., Liu, T., Xu, N., Lin, X., Wang, Y., & Wen, W. (2019, June). Feature distillation: Dnn-oriented jpeg compression against adversarial examples. In 2019 IEEE/CVF Conference on Computer Vision and Pattern Recognition (CVPR) (pp. 860-868). IEEE.
>
> [R3] Wong, E., Rice, L., & Kolter, J. Z. (2020). Fast is better than free: Revisiting adversarial training. arXiv preprint arXiv:2001.03994.
>
> [R4] Cohen, J., Rosenfeld, E., & Kolter, Z. (2019, May). Certified adversarial robustness via randomized smoothing. In international conference on machine learning (pp. 1310-1320). PMLR.
>
> [R5] Naseer, M., Khan, S., Hayat, M., Khan, F. S., & Porikli, F. (2020). A self-supervised approach for adversarial robustness. In Proceedings of the IEEE/CVF Conference on Computer Vision and Pattern Recognition (pp. 262-271).
>
> Q4. What formally is being preserved with the BD filters?
>
> (1) At the earlier layers, we change MaxPool, Conv, and AveragePool to MaxBlurPool, ConvBlurPool, and Blurpool, respectively (Lines 161-166 of the main paper). At the later layers, we introduce dilated convolutions in order to remove MaxPool or AveragePool.
>
> (2) Compared to the original filters, our BD filters preserve more low-level and low-frequency features (Please see the answer to Q1 and Q2). Compared to max pooling, which keeps only one feature in its receptive field, MaxBlurPool considers all features in its receptive field. Besides, compared to AveragePool, since we use a Blurpool with a larger kernel size, with the same stride, Blurpool can keep more low-level features.
>
>
>
>
> Q5. Threat model.
>
> (1) The threat model adopted in our work is consistent with the well-recognized work in this field (e.g., [6, 9]), which assumes that attackers know the training dataset of the target model.
>
> (2) We plan to study transfer attacks in more realistic settings in the future work.
>
>
>
> Q6. Implication of our method.
>
> Our method can be utilized to generate more transferable adversarial samples for adversarial training, which can better improve the robustness of a model.

---

> ### Author Response · Authors · 2023-08-18
> **Follow-up response**
>
> Dear Reviewer,
>
> Considering that the discussion phase is nearing to end, we are looking forward to your further feedback about our latest response. Do our responses fully address your concerns? Do you have any other comments? We would like to discuss with you in more detail. We greatly appreciate your time and feedback.
>
> Sincerely,
>
> Authors

---

> > ### Comment · Reviewer_1sbs · 2023-08-19
> >
> > Thank you to the authors for their thoughtful rebuttal. The simplicity of the approach and the results are compelling. Seeing results against various defenses to stop transfer attacks is also quite helpful. I agree with some of the other reviewers that some more thorough analysis of how BD is working so much better could be nice (hwet), and that there are some model specific / empirically guided settings (eif6), but given the strength of results I have increased my score nonetheless.

---

> > > ### Author Response · Authors · 2023-08-20
> > > **Thank you for the comment!**
> > >
> > > We are glad to hear that our response has addressed your concerns and you have increased your score. We will revise our manuscript according to the suggestions of all the reviewers. Thank you again for your time and encouraging comments!

---

### Official Review · Reviewer_VEyj · 2023-07-05

**Soundness:** 3 good
**Presentation:** 4 excellent
**Contribution:** 3 good
**Rating:** 6
**Confidence:** 3

**Summary:**

The paper proposes a novel transfer-based black box adversarial attack called Blurred-Dilated method. They authors consider the model modification approach and propose to reduce downsampling operations on the source model for the attack. They conduct extensive experimenting and compare with the previously proposed methods to show the superior transferrability of their generated adversarial examples.

**Strengths:**

1. Originality. The paper proposes a novel approach for the black-box attacks. I am not aware of other black-box attacks that focus on the forward propagation in the model.
2. Quality. The paper provides reasonable experimental support and justification of their proposed methodology.
3. Clarity. The paper is well-written and easy to follow.
4. Significance. Improving the transferability of the adversarial examples allows to raise awareness for the vulnerabilities of the models deployed in safety-critical domains.


**Weaknesses:**

The BD method proposed in this work seems to be specific for the CNN architecture and cannot be applied to other popular vision archictectures such as Vision Transformer [1]

[1] Dosovitsky et al. “An Image is Worth 16x16 Words: Transformers for Image Recognition at Scale”, ICLR 2021


**Questions:**

What is the effect of the introduced architectural elements on the model inference time?

**Limitations:**

Limitations and Broader Impact are adequately discussed in the Appendix

---

> ### Author Rebuttal · Authors · 2023-08-10
>
>
> Q1. Can the BD method be applied to other popular vision architectures such as Vision Transformer?
>
> We can first add CNN blocks to Vision Transformers as in (Dosovitsky et al. 2021). We can then apply our BD method to the CNN blocks in the hybrid  model to improve the transferability of adversarial attacks. We plan to extend our method to more architectures in the future work.
>
>
> Q2. What is the effect of the introduced architectural elements on the model inference time?
>
> We compared the inference time of Resnet50 and BD-Resnet50 for 300 iterations and obtained the average time. The result is that the inference time of Resnet50 is 7.85ms, and the inference time of BD-Resnet50 is 15.69ms. The above result is tested on a Nvidia Tesla T4.
>
> Compared to other methods, our method can still quickly generate adversarial samples with a higher attack success rate.

---

> > ### Comment · Reviewer_VEyj · 2023-08-14
> > **Reviewer's Response to the Rebuttal**
> >
> > Thanks for your rebuttal.
> >
> > Q1. Given the prevalence of Vision Transformers in the contemporary literature, it would be good to have quantitative results. I am not convinced that adding CNN blocks to the Vision Transformer and converting it into a hybrid model would improve the transferability of the attack. Looking forward to the results in your future work.
> >
> > Q2. Thanks for the results. Since inference time doubles after introducing blurred dilations, I find that this aspect should be clearly stated in the paper. Given that ResNet-50 has rather fast inference compared to other popular architectures, the overhead can be even more drastic for other families such as Vision Transformers.
> >
> > Given the rebuttal, I find that my original rating and confidence score are reasonable and I am not changing them. My confidence on the paper's impact and contribution remains not too high given the absence of results for the Vision Transformer architecture.

---

> > > ### Author Response · Authors · 2023-08-18
> > > **Thank you for the comment!**
> > >
> > > We will refine the paper accordingly in our final version. Many thanks for your valuable feedback!

---

### Official Review · Reviewer_hwet · 2023-07-07

**Soundness:** 3 good
**Presentation:** 3 good
**Contribution:** 2 fair
**Rating:** 5
**Confidence:** 2

**Summary:**

The paper investigates a modification to vision networks that causes adversarial examples generated by attacking them to be more transferable. In particular, the paper suggests using dilated convolutions + blurred downsampling, which the authors motivate as retaining a maximum amount of original feature information.

**Strengths:**

Overall, the paper seems reasonable and solid. It proposes a new approach (albeit with a somewhat handwavy motivation), shows that it works well in a variety of settings, and performs ablations to ensure that the components of their approach are necessary.

**Weaknesses:**

As alluded to above, although the paper is solid, it's not a particularly standout paper either. The motivation for why this approach is useful is quite vague and is not explored thoroughly, except for some light analysis of salience maps and the method itself is not very novel. The results, however, are quite good. Overall, i feel somewhat lukewarm on this paper.

**Questions:**

N/A

---

> ### Author Rebuttal · Authors · 2023-08-10
>
> Q1. The motivation of our approach.
>
> (1) We observed that: 1) The adversarial samples generated by ResNet usually have better transferability than those generated by other models. We think the reason may be that the skip connections of ResNet connect the lower-layer feature maps to the higher-layer feature maps, which helps to transfer more low-level features and reduce the information loss caused by downsampling. 2) Due to the large resolutions of the images on ImageNet, models need to do downsampling multiple times, resulting in a large amount of information loss. Besides, due to the structural differences between models, the discarded features are different. Therefore, preserving more low-level features, as ResNet does, can improve adversarial transferability. 3) On CIFAR-10/100, the adversarial transferability between models are usually better than that on ImageNet. Since the resolutions of the images on CIFAR-10/100 are smaller, there are fewer downsampling in the CIFAR-10/100 models. Therefore, different CIFAR-10/100 models all maintain more low-level features, making the generated adversarial sample easy to transfer.
>
> (2) Given the above observations, we think that employing a source model that can preserve more low-level features of an image during forward propagation can help to craft adversarial samples, which can more thoroughly and precisely destroy the low-level features of a clean image. Such adversarial samples are more transferable, since different models all use low-level features to extract high-level semantics and then make predictions. Since downsampling discards image features, which will reduce the details of an image (i.e., the low-level features) during forward propagation, we propose our Blurred-Dilated method to reduce downsampling in the original models, and preserve more low-level features of an image. Therefore, the proposed Blurred-Dilated method can generate more transferable adversarial samples.
>
>
>
>
> Q2. The novelty of our method.
>
> The novelty of our approach can be summarized as follows:
>
> (1) We find that existing transfer attacks based on model modification only focus on modifying the back propagation. Different from them, we are the first to consider forward propagation.
>
> (2) Our method is built upon a new perspective to improve adversarial transferability: keeping more low-level features during forward propagation. Besides, our method outperforms the state-of-the-art baselines by a significant margin. Therefore, our work sheds some light on studying the transferability of adversarial samples.

---

> ### Author Response · Authors · 2023-08-18
> **Follow-up response**
>
> Dear Reviewer,
>
> Considering that the discussion phase is nearing to end, we are looking forward to your further feedback about our latest response. Do our responses fully address your concerns? Do you have any other comments? We would like to discuss with you in more detail. We greatly appreciate your time and feedback.
>
> Sincerely,
>
> Authors

---

### Author Rebuttal · Authors · 2023-08-10

Dear Reviewers,

We sincerely appreciate all of your precious time and constructive comments. We are greatly encouraged by the positive comments on our work. We will carefully revise our manuscript by adding the suggested experimental comparisons, presenting more detailed explanations, and fixing the typos. We are looking forward to receiving your valuable feedback to further improve our work. Thank you for your time!

Sincerely,

Authors

---

> ### Comment · Area_Chair_qgxD · 2023-08-19
>
> Thank you for the response to all reviews, we will take the additional explanations and details into consideration for further discussions,
>
> Best regards,
> AC

---

### Decision · Program_Chairs · 2023-09-21

**Decision:**

Accept (poster)

**Comment:**

This paper proposes a novel method for transfer attacks, which is termed Blurred-Dilated. The focus of the method is on how to change a model architecture so that attacks on this model are more transferable to other models. The proposed Blurred-Dilated models use blurred downsampling and dilated convolutions in the network, motivated by the idea that important features are better preserved. The transferability is evaluated on ImageNet for a representative range of architectures.
The proposed approach is novel and its simplicity is appealing. While the reviewers point out that more evaluation of the Blurred-Dilated approach in terms of its theoretical motivation would be helpful, the empirical results in practical attack scenarios are most appealing. We encourage the authors to take the reviewers comments into account when preparing the final camera ready version of the paper.